# Evaluating the Perceived Health-Related Effectiveness of ‘The Daily Mile’ Initiative in Irish Primary Schools

**DOI:** 10.3390/healthcare12131284

**Published:** 2024-06-27

**Authors:** Luke Hanna, Con Burns, Cian O’Neill, Edward Coughlan

**Affiliations:** Department of Sport, Leisure and Childhood Studies, Munster Technological University, Bishopstown, T12P928 Cork, Ireland; con.burns@mtu.ie (C.B.); cian.oneill@mtu.ie (C.O.); edward.coughlan@mtu.ie (E.C.)

**Keywords:** The Daily Mile, Irish, primary school, health, physical activity, children, RE-AIM

## Abstract

Many Irish primary school children do not meet recommended physical activity (PA) guidelines. The Daily Mile (TDM) is a primary school initiative designed to increase children’s PA output. This study evaluates TDM’s perceived effect on Irish primary school children’s health-related metrics. A mixed-methods study, comprising two data collection phases, used the RE-AIM framework to evaluate TDM. Phase One involved teachers, principals and classroom assistants (*n* = 191) from TDM-registered schools completing a questionnaire. Two TDM-registered schools participated in Phase Two, where (i) interviews were conducted with each principal, (ii) a teacher sub-sample (*n* = 4) participated in a focus group, and (iii) a child sub-sample (*n* = 14) participated in separate focus groups. TDM was perceived to positively impact the markers of children’s health, including their PA behaviour, physical fitness and social well-being. Inclement weather (66.5%) and a lack of time (56.5%) were reported as the top-ranked implementation barriers. Moreover, TDM’s repetitive nature often left children feeling disinterested and resulted in some teachers modifying the initiative to maintain engagement levels. Maintaining the long-term implementation of TDM and its health benefits in primary schools may require bespoke amendments to the original format to preserve children’s engagement.

## 1. Introduction

Habitual physical activity (PA) supports growth and development, and it has positive effects on children’s physical, cognitive and psychosocial health [1]. PA and physical fitness have a positive relationship and are associated with improved executive function and academic performance in children [1]. Regular PA during childhood is associated with a reduced risk of developing chronic illnesses such as obesity, type 2 diabetes and cardiovascular disease [2]. Furthermore, habitual PA during childhood directly impacts positive health outcomes later in life, and this positive PA behaviour has been shown to continue from childhood into adulthood [3]. Children are recommended to engage in at least 60 min of moderate-to-vigorous physical activity (MVPA) every day [4,5]. Woods et al. reported that a mere 23% of Irish primary school children achieve this daily MVPA recommendation [6]. Notably, however, this represents an improvement from an earlier study by the same authors, where only 17% of Irish primary school children were considered to meet the daily PA guidelines [7]. Primary school is an appropriate setting to target improving PA behaviour as children must attend for nearly six hours per day for approximately half of a typical calendar [8]. Moreover, children’s regular adherence to a school-based PA intervention has the potential to positively impact a variety of health-related metrics [9].

The Daily Mile (TDM) is a PA initiative that was developed in a Scottish primary school in 2012 to tackle their children’s perceived lack of fitness [10]. The dissemination of TDM from the school where it initiated has resulted in the initiative being adopted by primary schools across 86 different countries, with over 1100 Irish schools registered as TDM participants [11]. Several research studies have been published that suggest TDM participation has the potential to positively impact children’s health-related metrics, namely cardiorespiratory fitness (CRF) and PA behaviour [12,13,14,15,16,17,18,19,20,21]. However, the level of implementation was not measured in many of the studies assessing the longitudinal relationship between TDM and various health-related outcomes [15,20,22]. The dose of PA is often categorised based on the accumulation of exercise factors such as type, duration and intensity, and the precise monitoring of the dose exposed to participants is crucial for accurately attributing outcome effects to an intervention [23,24]. Subsequently, the results of TDM-related studies that did not measure the dose received, must be interpreted cautiously.

Previous research studies have evaluated metrics related to TDM implementation in primary school contexts across geographical locations such as Scotland, England, Wales and Northern Ireland [10,20,25,26]. The continued implementation of TDM has positively affected and has been facilitated by the initiative’s perceived impact on physical outcomes such as physical fitness, physical literacy and PA patterns [10,17,19,20,25,27,28]. Moreover, TDM has been attributed as having a positive social effect for children, and it reportedly influences the development of a strong social rapport between children and teachers [17,19,20,25,27]. Teachers’ encouragement, enthusiasm and participation in the implementation process enhances children’s motivation and engagement with TDM [17,20,26,27,28]. Conversely, time constraints associated with delivering TDM and the demands of the existing school curriculum are critical barriers to implementation [17,20,25,26,27,28]. Consequently, providing teachers with autonomy and flexibility over the implementation and modification of TDM’s components appears to facilitate the successful delivery of the initiative [10,17,20]. Moreover, research suggests that schools and teachers regularly adapt and evolve TDM’s original format to overcome common implementation barriers and to cater to the needs and preferences of participating children [17,20,25,27,28].

Researchers should adopt a well-rounded approach when measuring the success of TDM adoption, implementation and maintenance to ensure all relevant stakeholders continue to perceive it as a beneficial activity for primary school children’s health. No research evaluating the implementation and perceived health-related effectiveness of TDM within an Irish primary school setting has been conducted previously. Accordingly, the impact an Irish school’s classification (i.e., DEIS vs. Non-DEIS) or location (i.e., urban vs. rural) have on TDM’s perceived health-related effectiveness have not been previously measured. DEIS schools in Ireland receive additional government funding and resources to ensure that the educational needs of children from disadvantaged communities are met [29].

This study adopted a holistic approach and used the RE-AIM (Reach, Effectiveness, Adoption, Implementation and Maintenance) framework [30] to bridge the gap in evidence by analysing the perceptions teachers, principals and children have of TDM within an Irish primary school context. RE-AIM attempts to determine the level of external validity associated with public health interventions, subsequently identifying which populations it works best for and how it can be delivered successfully [31]. The RE-AIM framework’s components used to facilitate the evaluation process are as follows: reach, which identifies the complete number, and the percentage and representativeness of the target population who are willing to participate in an initiative. Effectiveness refers to the initiative’s impact on specified outcomes. Adoption is the complete number, proportion and representativeness of individuals in a setting who are willing to commence delivering the initiative. Implementation refers to the level of fidelity with which the pre-determined components of the initiative are delivered. Maintenance relates to the initiative’s sustainability [32]. In support, the RE-AIM framework has been used to evaluate the implementation and effectiveness of school-based PA interventions [33,34,35,36,37].

Therefore, the purpose of this study was to achieve the following: (1) use the RE-AIM framework to evaluate the perceived health-related effectiveness of TDM in Irish primary school settings, (2) identify the facilitators and barriers associated with the successful implementation of TDM in Irish primary schools and (3) evaluate the adoption, implementation and long-term uptake of TDM in Irish primary schools.

## 2. Materials and Methods

### 2.1. Study Design

A mixed-methods approach, involving two phases of data collection, was adopted to satisfy the research objectives. The first phase involved the cross-sectional data collection of quantitative information using a web-based questionnaire. The questionnaire was completed over a 2-month period, from December 2020 to January 2021, by teachers, principals and staff members working in primary schools experienced with implementing TDM. The second phase involved the cross-sectional collection of qualitative data from relevant TDM stakeholders (i.e., principals, teachers and children) through the administration of semi-structured interviews and focus groups over a 2-month period, from May 2021 to June 2021, 4-months after the completion of Phase One. TDM was implemented concurrently with the scheduled physical education (PE) classes in the primary schools of this study’s participants.

### 2.2. Questionnaire Data Collection

The questionnaire was distributed to all Irish primary schools registered as TDM participants using a convenience sampling method. The database of all Irish primary school participants was searched to retrieve the contact details of each school’s lead TDM coordinator. These personnel were emailed and invited to participate in the questionnaire. They were also requested to circulate the questionnaire to all staff within their school. This process was coordinated by Athletics Ireland, i.e., the promoters and coordinators of TDM in Ireland.

The questionnaire was developed and based on the RE-AIM framework [32]. Reach was evaluated by calculating the percentage of Irish primary schools that were registered as TDM participants. Effectiveness was measured by examining the perceived impact that TDM had on children’s health-related metrics. Adoption was measured by examining the perceptions related to the delivery of and participation in TDM. Implementation was evaluated by examining the impact potential facilitators and barriers had on the delivery process. Maintenance was evaluated by exploring the participants’ commitment to continuing the implementation of TDM in their schools.

The questionnaire was piloted for reliability and validity by primary school principals (*n* = 2) and teachers (*n* = 6) (Appendix A). The questionnaire was hosted on an online survey platform known as LimeSurvey, version 3.0 (Heidelberg, Germany) [38]. Skip logic was implemented to ensure participants were only shown questions relevant to them and their experiences of TDM. The mean completion time recorded by the survey platform was 6.5 min.

### 2.3. Focus Group and Interview Data Collection

Qualitative data were collected from principals, teachers and children from participating schools through the administration of interviews and focus groups. Convenience sampling was used to recruit primary schools with experience of implementing TDM from a pre-existing database of TDM participants. The primary researcher subsequently contacted each school and invited them to participate in this study. Two primary schools from southern Ireland, both classified as Non-Delivering Equality of Opportunity in Schools (Non-DEIS), accepted this invitation and agreed to participate. One participating school (School A) is located within an ‘independent urban town’, and the other participating school (School B) is located within a ‘rural area with moderate urban influence’ [39]. This mix of schools facilitated the evaluation of the effect a school’s location (i.e., urban vs. rural) has on the TDM’s perceived effectiveness. The interviews and focus groups were conducted on the online Zoom version 4.6.19178.0323 platform (San Jose, CA, USA) by the primary researcher due to restricted access to Irish primary schools during the COVID-19 pandemic [40].

Interviews were conducted with each participating school’s principal. One focus group was held that involved the participation of teachers (*n* = 4) from both schools. A focus group was used to gain a further insight into the implementation and perceived effectiveness of TDM as they allow individuals to provide further commentary on other participants’ opinions and perceptions. The principal interviews and teacher focus groups followed a similar semi-structured topic guide (Appendix A). The questions presented in previous research, which evaluated TDM’s implementation process, supported the design and development of the topic guides followed in the principal interviews and teacher focus group that were used in this study [10,20]. The primary researcher used additional prompting questions if the original question did not elicit clear and detailed responses from the participants. The questions and prompts were piloted beforehand with a school principal in a 1:1 interview and with primary school teachers (*n* = 3) in a focus group held on the Zoom platform.

Two focus groups that followed a different format from the teacher focus group and principal 1:1 interviews were held with children from each participating school (Appendix A). Children from the two oldest-aged class groups (aged 10–12 years old) in both schools participated as it was perceived they would provide more detailed information about their experiences and perceptions of TDM than children from the youngest-aged class groups. Children were instructed to draw and write about what participating in TDM means to them. Children were then randomly selected by their teacher to participate in a focus group held on the Zoom platform to further discuss their experiences and perceptions of TDM. This ‘Write & Draw’ technique, developed by Williams et al. [41], was successfully utilised by participants in a previous study to evaluate a school-based PA and nutrition intervention [36]. Two separate focus groups were held for the participating children from each school. Six children from School A participated in the first child-centred focus group and eight children from School B participated in the second child-centred focus group. The questions and prompts used in the child-centred focus groups were piloted with children *(n* = 2) on the Zoom platform in March 2021.

### 2.4. Data Analysis

Data were analysed using the IBM Statistical Package for the Social Sciences (SPSS), version 26.0 (Chicago, IL, USA). The imported data were cleaned to ensure there were no errors across the variables. Data were analysed by conducing relative frequencies across various variables. Separate analyses were conducted for school classification (i.e., DEIS/Non-DEIS) and location (i.e., urban/rural). Kolmogorov–Smirnov and Shapiro–Wilk goodness-of-fit tests determined that the data were not normally distributed. Subsequently, non-parametric tests were conducted to analyse the relationship between independent and dependent variables. Chi-squared tests for independence and Mann–Whitney U-Tests were administered to identify the effect of a school’s classification and location.

This study adopted an interpretative approach, using thematic analysis to analyse the qualitative data collected [42]. This involved the primary researcher identifying interesting features (i.e., codes) within the data before grouping these into broader patterns (i.e., themes) for further analysis [42]. These themes were organised according to the RE-AIM element deemed most relevant to each theme’s topic. The analysis was facilitated by transcribing the dialogue verbatim with syntax from the digital recordings onto a Microsoft Word document. The initial stage involved the primary researcher familiarising himself with the qualitative data. During this phase, the primary researcher took notes on common topics raised and discussed throughout the interviews and focus groups, which then facilitated the generation of a coding framework. The transcribed scripts were read and re-read, with relevant data coded under the appropriate section of the established framework. The identified codes were then grouped under higher-order themes and sub-themes. These themes were reviewed to ensure the included coded data accurately reflected the nature of the theme it was grouped under. Microsoft Word was used to store identified themes under appropriate headings. The resulting findings are presented to support the quantitative questionnaire data collected in the results section below.

## 3. Results

A sample of females (84.3%) and males (15.7%), which reflects a representation of the sex of teachers working in Irish primary schools, completed the questionnaire [43]. Participation of individuals from DEIS (20.9%) and Non-DEIS (79.1%) schools was also somewhat illustrative of the classification of Irish primary schools [29]. The response rate was 54.5% in rural and 45.5% in urban schools. Similarly, there was a comparable proportion of responses collected from participants in the younger- (<40 years old = 49.7%) and older-aged (≥40 years old = 50.3%) categories. Class teachers (58.6%) and co-educational primary schools (87.4%) were well represented in the questionnaire data (Table 1 and Appendix A Appendix A). Principals and vice principals who taught children and classes (i.e., teaching principals and teaching vice principals), as well as principals and vice principals who did not teach a class (i.e., walking principal and walking vice principal), were instructed to distinguish themselves from each other when answering a question relating to their occupation in the questionnaire (Table 1).

### 3.1. Reach

In Ireland, there are 1,169 schools from a total of 3,300 that are registered as TDM participants [11,44], equating to a reach percentage of 35%.

### 3.2. Effectiveness

TDM was found by the questionnaire participants to positively impact a range of potential benefits associated with children’s health (Table 2).

A Mann–Whitney U test was used to conduct a sub-group analysis, which revealed that a significantly higher percentage of rural school participants ‘agreed’ or ’strongly agreed’ that TDM positively impacted children’s health and well-being (97.1% vs. 85.1%; *p* = 0.01), physical fitness (93.3% vs. 80.5%; *p* = 0.02) and movement proficiency (87.5% vs. 77%; *p* = 0.01) when compared to urban school participants. There was no statistical difference found between the schools’ classification (i.e., DEIS vs. Non-DEIS) and any benefits potentially associated with TDM. Illustrative quotes from an analysis of the qualitative data relating to TDM’s health-related effectiveness, with a focus on the most frequent sub-themes, are presented in Table 3.

### 3.3. Adoption

Questionnaire data revealed positive perceptions about adopting TDM among teachers and children in participating schools. Most participating teachers (96.7%) reported feeling ‘positive’ or ‘very positive’ about adopting TDM into their teaching schedule. A total of 2.6% of participating teachers reported being ‘neither positive nor negative’ about adopting TDM into their teaching schedule, with only 0.7% reporting negative feelings towards adoption. Furthermore, 94.8% of teachers perceived that children felt either ‘positive’ or ‘very positive’ about participating in TDM. A total of 3.9% of teachers perceived children felt ‘neither positive nor negative’, with 1.3% perceiving that children felt ‘negative’ about TDM participation. No significant differences were found across school location (i.e., urban/rural) and classification (i.e., DEIS/Non-DEIS) for perceptions associated with the adoption of TDM in participating schools. Illustrative quotes from the analysis of qualitative data relating to the adoption of TDM are presented in Table 4.

### 3.4. Implementation

Overall, a high percentage of teachers (90.5%) reported that they implement TDM with their class, whereas 9.5% of teachers reported that they did not implement the initiative. Among those teachers who reported implementing TDM, 90.8% delivered the initiative three or more days every week (Figure 1).

Furthermore, 92.2% of teachers implementing TDM reported that they felt ‘confident’ or ‘very confident’ when doing so. A total of 7.8% of TDM implementors responded that they were ‘neither confident nor lacking in confidence’ when delivering the initiative. The ‘school yard’ was identified as the location most frequently used to deliver TDM by teachers (79.7%), followed by a ‘grass pitch’ (7.2%), an ‘external facility’ (5.2%) and a ‘running track’ (3.3%).

Approximately 1 in 6 implementors of TDM (16.3%) reported that children in their class used various PA methods other than walking or running when participating in TDM. These TDM implementors described in a follow-up question that children engaged in various games (e.g., dodgeball, capture the flag, tag chase, basketball, gymnastics, etc.) and alternative PA methods (e.g., skipping, hopping, burpees, side-stepping, etc.) during the delivery of a TDM session. Conversely, 83.7% of teachers responded that children in their class do not use PA methods other than walking or running when participating in TDM. More than half of the teachers reported regularly participating in TDM with their class (58.8%), with 25.5% of teachers ‘often’ participating and 33.3% ‘always’ participating (Figure 2).

From a range of barriers presented, ‘inclement weather’ (66.5%) and ‘a lack of time during a school day’ (56.5%) emerged as the most common barriers that the questionnaire participants associated with implementing TDM (Figure 3).

A chi-squared test revealed that rural school teachers were significantly more likely to implement TDM than urban school teachers (95.6% vs. 84.4%; *p* = 0.002). A chi-squared test also found a significant association (*p* = 0.01) between the schools’ classification and the location where teachers usually implement TDM. In support, a higher percentage of DEIS teachers (13.3%) usually implemented TDM on a running track when compared to teachers from Non-DEIS schools (0.8%). No statistical associations were identified between the schools’ location (i.e., urban/rural) for the usual implementation location, weekly frequency of delivery or confidence levels associated with implementing TDM. No statistical difference in the schools’ classification (i.e., DEIS/Non-DEIS) were exhibited for the implementation, frequency of delivery or confidence levels associated with implementing TDM. No statistical differences were observed between the schools’ location or classification for the PA methods implemented during TDM, the teachers’ level of participation in the initiative or potential implementation barriers. Illustrative quotes from an analysis of the qualitative data relating to the implementation of TDM with a focus on the most frequent sub-themes, namely barriers and facilitators, are presented in Table 5.

### 3.5. Maintenance

Almost all questionnaire participants (99%) would recommend TDM to other primary schools. Furthermore, 86.4% of staff members would like to see their school commit to TDM for at least another year or more (Figure 4).

No significant differences were found across school location (i.e., urban/rural) or school classification (i.e., DEIS/Non-DEIS) for results relating to the maintenance of TDM in primary schools. Illustrative quotes from an analysis of the qualitative data relating to the long-term uptake of TDM are presented in Table 6.

## 4. Discussion

TDM has been adopted and implemented in over 1100 Irish primary schools [11], which equates to over 1 in 3 Irish primary schools registered as TDM participants [44]. The current study reported that teachers and staff in Irish schools implementing TDM were positively disposed to the initiative, recognising its beneficial impact on children’s health and well-being. These findings align with previous research reporting the positive impacts of TDM on children’s social health and the markers of physical health such as CRF, PA behaviour and body composition [13,14,15,16,18,20,21,45,46]. Moreover, previous research displayed how TDM promotes and encourages social interaction between children, while also developing the social rapport between teachers and children [17,19,20,25,27]. In support, associated perceived benefits have been shown to influence the adoption, implementation and long-term uptake of PA initiatives such as TDM [47]. Furthermore, the results suggest that participating in TDM positively affects cognitive processes such as concentration and behaviour, which is similar to previous studies that have also reported TDM as having a positive impact on aspects of children’s executive function [12,46]. Conversely, Morris et al. and Martins et al. reported that TDM did not significantly impact children’s executive function [21,48]. Moreover, Booth et al. reported that children with no experience of TDM participation performed significantly better on a visual–spatial working memory test compared to children who had participated in TDM for at least one year [45]. In addition, Hatch et al. reported no significant association between TDM and children’s executive functions, but their results did suggest that TDM may positively impact accuracy performance in tests measuring inhibition and visual working memory [19]. The robust methodological design and control of confounding variables such as dietary intake by Hatch et al. and Dring et al. [19,46] suggests participating in TDM may have the potential to impact elements of children’s cognition. Future research should attempt to address and resolve the conflicting evidence presented in previous studies, as well as further explore TDM’s chronic impact on children’s cognition. The significantly higher implementation rate found among rural school teachers may have contributed to the greater agreement among rural staff regarding TDM’s potential health benefits when compared to urban school staff.

TDM stakeholders who participated in this study generally perceived the adoption of TDM as positive and were committed to sustaining the long-term implementation of the initiative. Additionally, selecting someone to effectively lead and co-ordinate TDM was identified as an important adoption and implementation facilitator. This aligns with previous research highlighting the crucial role a coordinator plays in adopting, implementing and maintaining a PA initiative [47,49,50]. According to Cassar et al. [47], school policy and a shared decision-making process among staff facilitates the adoption of school-based PA initiatives such as TDM. Similarly, the results of this study suggest that introducing a formal school policy facilitates the comfortable adoption of PA initiatives like TDM. However, the results also demonstrate that affording teachers with flexibility and autonomy can help smoothly integrate TDM into a school’s daily routine.

The results in this study suggest that TDM is regularly implemented by teachers, which is consistent with previous studies reporting similar rates of TDM implementation [14,17,25]. According to Durlak and DuPre [49], a 60% rate of program implementation regularly results in positive outcome effects. The finding presented in this study suggests the rate of TDM implementation in Irish primary schools can induce positive health-related changes. Moreover, the quantitative data collected in this study suggest a higher rate of TDM implementation in rural primary schools than in urban schools. As per Loucaides et al. [51], rural primary school children in Cyprus have significantly more neighbourhood space available to engage in PA than urban primary school children. Subsequently, it may be presumed that Irish rural primary schools have more available space than their urban counterparts, which facilitates the successful implementation of PA initiatives like TDM.

The implementation and long-term uptake of PA initiatives such as TDM requires the assistance of a supportive school climate [9,49,50]. Comparably, findings from this study and other TDM-related studies [10,17,20,25,26,28] demonstrate the impact that teachers’ involvement and participation in TDM have on children’s motivation and engagement with the initiative. Similar to previous research that analysed TDM’s implementation process [17,20,25,27,28], time constraints associated with delivering TDM in combination with classroom curriculum pressures emerged as potential implementation barriers. Comparably, Naylor et al. identified time constraints as the primary barrier to implementing school-based PA initiatives [9]. According to Ryde et al. and Harris et al. [10,17], the flexibility of TDM supports teachers’ autonomy and limits the negative impact that implementation has on children’s learning time.

This study’s data analysis revealed how weather and school facilities can prevent TDM delivery, aligning with previous studies that identified common TDM implementation barriers [10,17,20,25,27,28]. DEIS schools in Ireland receive additional government funding and resources [29], potentially explaining why a significantly higher percentage of Irish teachers from DEIS schools reported implementing TDM on a running track when compared to teachers from Non-DEIS schools. Accordingly, DEIS schools’ facilities may support greater implementation rates and long-term uptake of PA initiatives such as TDM.

This study highlights how TDM’s repetitive nature can leave children feeling bored and disinterested, potentially limiting the initiative’s sustainability in Irish primary schools. Adapting and modifying elements of the original format and implementing variations of TDM were exhibited to increase children’s engagement with the initiative. Comparably, previous studies illustrate the importance of developing and evolving core components of TDM to align with the needs and desires of each participating school [10,17,20,25,27,28]. According to child participants in the research of Hatch et al. [19], incorporating a variety of PA options and introducing a competitive element are required to overcome the barrier associated with TDM’s repetitive and boring nature. In addition, findings from the studies by Breslin et al. and Ram et al. suggest that many primary schools in Northern Ireland and London, England, do not implement the core principles of TDM with fidelity [26,52]. Previous research demonstrates how adaptable PA interventions can promote and influence greater success rates of implementation in primary schools [47,49]. In support, no school-based intervention analysed in the research of Herlitz et al. was sustained in its entirety [50]. This suggests developing and progressing the original formats of PA initiatives like TDM are necessary to ensure their long-term uptake in primary schools.

Identifying the core elements of an initiative linked to positive outcomes helps distinguish which components should be implemented faithfully and which can be adapted to meet participants’ needs and preferences [49]. Future researchers should strive to determine which elements of TDM are related to positive health-related outcomes. Measuring the level of adherence, dose received, implementation quality, participant engagement and the level of adaptations made to the initiative are required in future studies examining fidelity to TDM’s original format [53]. Moreover, it is necessary to identify TDM’s features that negatively affect children’s engagement as these can impact the initiative’s sustained implementation and associated health benefits in Irish primary schools. As recommended by Durlak and DuPre [49], future studies should monitor and record the adaptations made to PA initiatives to understand their impact on implementation and sustainability.

A wide range of Irish primary school stakeholders participated in this mixed-methods study, which included a questionnaire survey, interviews and focus groups. The data were collected with the purpose of evaluating TDM’s perceived health-related effectiveness in Irish primary schools. The perceived effectiveness of TDM on markers of children’s health in Irish primary school settings was accurately represented through the collection of questionnaire data from schools located in every Irish province. Furthermore, urban and rural primary schools were well represented within each data collection phase, facilitating an accurate prediction of the impact a school’s location has on the implementation and perceived effectiveness of TDM.

Convenient sampling methods were used to recruit schools and participants for both phases of the study. These methods are unlikely to represent schools disinterested in TDM; thus, implementation barriers may be underdeclared as a result. Moreover, this study was unable to recruit a DEIS primary school to participate in the second phase of data collection. Consequently, the qualitative methods were limited in their ability to further explore the impact a school’s classification (i.e., DEIS/Non-DEIS) has on the implementation and perceived effectiveness of TDM. Additionally, direct observation of TDM implementation was not feasible due to primary schools restricting entry to external visitors during a global pandemic. Accordingly, the data gathered may have favoured a positive outcome for TDM as study participants were likely to support implementing and maintaining the initiative in their school’s routine.

Elements of TDM may need to be developed and evolved to ensure the continued implementation of this initiative and the subsequent health benefits in Irish primary schools. Future evaluations of TDM should identify adaptations associated with positive health-related outcomes, increased engagement levels and successful implementation in primary school settings.

## 5. Conclusions

A large proportion of Irish primary schools regularly participate in TDM, which appears to positively impact various aspects of children’s health and well-being. The implementation process affects children’s participation, enjoyment and engagement levels, as well as the long-term uptake of TDM within schools. Teacher and staff buy-in and support for TDM appear crucial to overcoming implementation barriers, such as inclement weather and concerns about TDM’s impact on learning time. Moreover, the long-term sustainability of children’s positive engagement with TDM may require bespoke amendments to the original format to stimulate interest, optimise enjoyment and maximise health-related outcomes.

## Figures and Tables

**Figure 1 healthcare-12-01284-f001:**
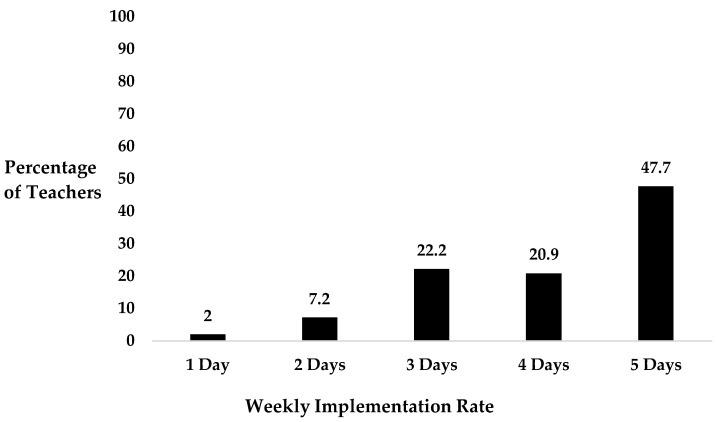
Mean weekly TDM implementation rate.

**Figure 2 healthcare-12-01284-f002:**
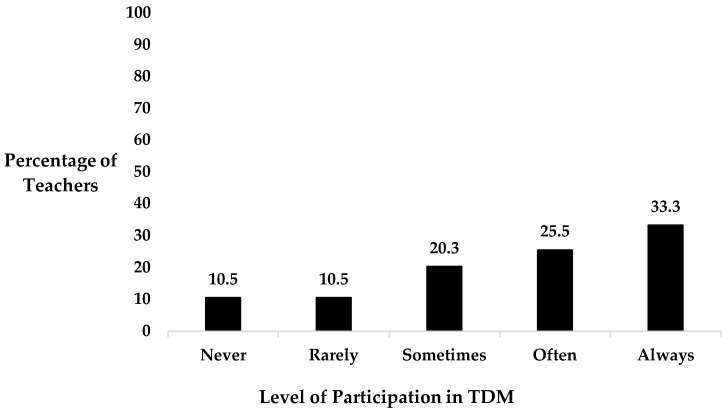
Breakdown of teachers’ level of participation in TDM with their class.

**Figure 3 healthcare-12-01284-f003:**
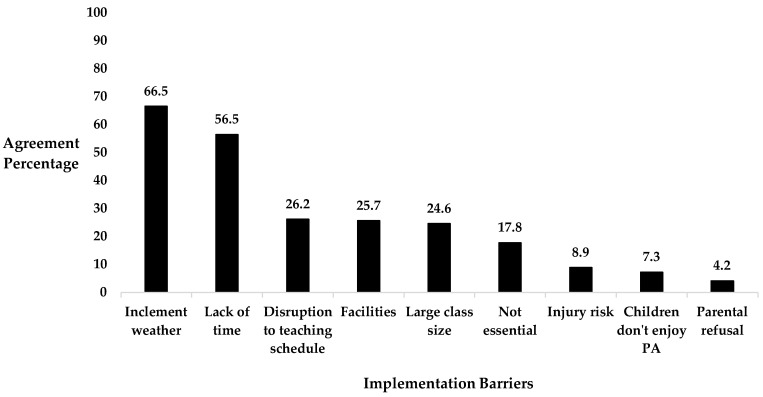
Percentage of questionnaire participants who ‘strongly agreed’ or ‘agreed’ that the suggested barriers impacted TDM implementation.

**Figure 4 healthcare-12-01284-f004:**
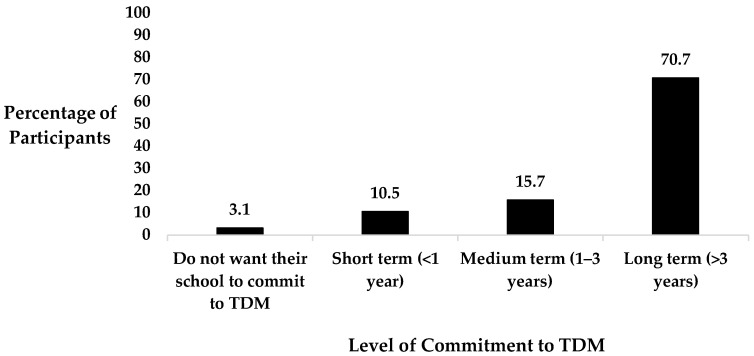
Breakdown of the questionnaire participants’ feelings towards their school’s commitment to TDM.

**Table 1 healthcare-12-01284-t001:** Breakdown of participants’ occupations at their schools.

Occupation	N	%
Walking Principal	11	5.8
Teaching Principal	21	11
Walking Vice Principal	0	0
Teaching Vice Principal	20	10.5
Class Teacher	112	58.6
Resource Teacher	16	8.4
Special Needs Assistant	3	1.6
Other *	8	4.2

* Questionnaire participants were instructed to choose ‘other’ if they did not occupy a function displayed in the question.

**Table 2 healthcare-12-01284-t002:** The ratings of the participants regarding their agreement with potential TDM benefits based on their experiences with the PA initiative.

Perceived Benefits of The Daily Mile (TDM)	Strongly Disagree or Disagree (%)	Undecided (%)	Strongly Agree or Agree (%)
Positively impacts children’s health and well-being	4.2%	4.2%	91.6%
Positively impacts children’s attitudes towards physical activity (PA)	4.2%	4.7%	91.1%
Positively impacts children’s participation in PA	4.2%	11.5%	84.3%
Positively impacts children’s physical fitness	4.2%	8.4%	87.4%
Positively impacts children’s movement proficiency	3.1%	14.1%	82.7%
Reduces children’s sedentary behaviour during school	2.1%	9.4%	88.5%
Increases the opportunity for social interaction between children	3.7%	14.7%	81.7%
Positively impacts children’s attentiveness and concentration directly after a TDM session	3.7%	14.7%	81.7%
Positively impacts children’s attentiveness and concentration across a school day	3.7%	20.4%	75.9%

**Table 3 healthcare-12-01284-t003:** Sub-categories and quotes from the interviews and focus groups that represent the health-related effectiveness of TDM.

Sub-Theme of Effectiveness	Topic of Quote	Quote
Physical health	TDM’s impact on less athletic children.	*‘The ones who could not do one lap at the beginning. They’re the ones that are making the most progress’* (Teacher, School A).
	TDM helps to minimise the prevalence of child obesity.	*‘In certain homes, where you may not have the same support or they may not be taking part, this is their exercise. This is what we do every day and when they were missing out on a mile a day, it’s amazing the difference it made to their physical size’* (Principal, School B).
Social health	TDM fosters the development of friendships.	*‘There are some people that normally just would not mix. They would be in different circles or different sports, but sometimes they’re just at that pace, the same pace as someone that’s running beside them and you’d often see them chatting away to each other. It has actually helped to develop a good few friendships’* (Teacher, School B).
Cognitive health	Classroom impact.	*‘When you’re gone out to the fresh air and you come back in, you’re ready to sit down and start your work again’* (Child, School B).
	Behavioural impact.	*‘Since we brought in TDM, our discipline problems are not huge, but our discipline problems have plummeted’* (Principal, School B).

**Table 4 healthcare-12-01284-t004:** Quotes from the interviews and focus groups that represent perceptions related to the adoption of TDM.

Topic of Quote	Quote
Differentiated approaches for integrating TDM into teaching schedules.	*‘The teacher decides what time is most appropriate. I do not think we had to sit down and say is everybody doing TDM? Are we all on board? That never happened at a staff meeting. We said we’re introducing TDM and we’d love everybody to be part of this and that’s exactly what happened. Therefore, there was no formal school policy that we all have to do this now’* (Principal, School A).
	*‘They have a timetable so unless the weather is absolutely shocking, they have their 10-15 min slot and it starts at about 10:15 in the day and it continues throughout’* (Principal, School B).
The role of a TDM coordinator.	*‘If there’s a hassle, if there’s a problem here he’ll come to myself, you know if there’s something huge, but it’s great we work away and thanks be to God we have had no issues. You want it to be successful, so if there was a problem [name of coordinator] can work with it’* (Principal, School B).
The impact of a school’s PA culture and uniform policy.	*‘Everybody is dressed appropriately because we are wearing a school track suit every day at school. We do not have a formal school uniform. 99% of the children are wearing runners coming to school, you know they’re all wearing appropriate footwear coming to school’* (Principal, School A).

**Table 5 healthcare-12-01284-t005:** Sub-themes and quotes from the interviews and focus groups that represent TDM’s implementation process.

Sub-Themes of Implementation	Topic of Quote	Quote
Barriers	Inclement weather and inadequate facilities.	*‘I know when we were doing runs in the grass and stuff like that, when the pitch is out of use there for the months of December and November, you’re not really running it then’* (Teacher, School B).
	Time constraints associated with a congested curriculum.	*‘It can take a bit of time. You’re talking the bones of 25 min, half an hour a day, which is a lot to be giving towards it when you have other items you’re supposed to be covering as well’* (Teacher, School A).
	TDM’s repetitive nature.	*‘I think it’s good but it’s sort of a repetitive thing to do the same four laps every day of the same pitch. It’s not like other sports like soccer, basketball or hurling where it’s always different each day’* (Child, School A).
Facilitators	Social rapport between teachers and children.	*‘You could be chatting away to them as well. I find that’s nice because I find it so busy during the days you do not get a chance to talk to half of them with stuff that’s going on, and some of them like chatting away to you and jogging away’* (Teacher, School B).
	TDM’s inclusive nature.	*‘What it helped greatly actually was children who may have autism, who may find it hard to take part in games that are not organized. Now they’d something to do so, they were going around with a pal or going around with somebody else’* (Principal, School B).
	Teachers participating in TDM.	*‘Yeah I do it, not every day but maybe half the time I jog around and I tend to stay around the middle or towards the back and kind of give them a bit of a G-up because the other ones at the front are tearing off ahead’* (Teacher, School A).
	Adapting and varying how TDM is implemented.	*‘Some people have a different twist. Today’s Thursday and there were some classes doing relays. Some of the junior classes might start it off where they’ll just walk and then you can get them jogging you know, walk one and then jog one. The older classes, I’m looking at sixth class there and there’s some guys in there just doing rounds. Some teachers might tie it in with a homework pass or something like that, you know every so often get your laps up this week, or we do a marathon or something like this and every so often, you do have to give an incentive’* (Principal, School B).
	Addition of a competitive element.	*‘The competitive nature for me. I find a lot of them, like there’s a couple of boys and girls who would be very involved in their sport and they’d be running and they’d be chatting away but they do not want to finish behind someone else. I know that they enjoy it, but they still take the competitive part of it very seriously as well’* (Teacher, School B).

**Table 6 healthcare-12-01284-t006:** Quotes from the interviews and focus groups related to the long-term uptake of TDM in primary schools.

Theme of Quote	Quote
Prioritising the personal development of each child.	*‘As long as the emphasis is on self-improvement, they will not get too caught up on who is the fastest or who is the slowest so they’re all making their own improvement. I’ll definitely keep it up’* (Teacher, School A).
Recognising the benefits of TDM participation.	*‘I think we should keep on participating because even though sometimes when we might not want to run, the fresh air and just running with your friends always helps the mind even if you do not really feel like running’* (Child, School B).
Recognition of a school’s TDM commitment.	*‘We’d love to commit to implementing it and we’d love to have a Daily Mile flag outside the school at some point. Maybe there should be some sort of recognition. I know we are involved, and we’re registered but maybe there should be some sort of recognition, somewhere where people say this is a Daily Mile school* (Principal, School A).

## Data Availability

The data presented in this study are available on request from the corresponding authors as the data contain sensitive information that cannot be shared openly due to data protection regulations.

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
