# Peer review of "Evaluating the Perceived Health-Related Effectiveness of ‘The Daily Mile’ Initiative in Irish Primary Schools"

_healthcare, 2024, doi:10.3390/healthcare12131284_

Round 1
Reviewer 1 Report
Comments and Suggestions for Authors
This is a well written paper and an interesting study of an implemented initiative.
In what years, was phase 1 and phase 2 data collected. What was the time period between these 2 phases of data collection. Did these schools also have physical education? Was the TDM in lieu of PE classes? If they also had phys ed classes this may have impacted implementation.
Reviewer 2 Report
Comments and Suggestions for Authors
The authors address an important topic: increasing physical activity, especially among children. This paper helps to evaluate an established framework and identify factors essential (and those interfering with) implementation. These factors may help to raise awareness of how to be more effective in this endeavor.
I only have minor comments:
line 288: "Chi-Square" should be chi-squared. Greek letters are not capitalized, and while this is commonly referred to as "square", it is more technically correct to state "squared".
Just before the conclusion, I suggest adding a paragraph detailing the limitations of the study.
Reviewer 3 Report
Comments and Suggestions for Authors
Dear Authors,
Thank you very much for the opportunity to read your engaging and very well-prepared manuscript. Physical activity is essential to people's lives at every stage, and children in particular. The Irish schools initiative fits perfectly into this trend, and the Authors have accurately diagnosed the weaknesses of this project.
As I wrote earlier, the article was carefully prepared regarding content and graphics.
I only have reservations about the abstract itself. As abstracts are also published separately from the article, citing literature in the abstract is not allowed. It is also not recommended to include abbreviations whose meaning we develop in the article (I am referring to the abbreviation RE-AIM).
As the subject matter is popular in Ireland, I found out by typing The Daily Mile in the PubMed database that recent articles can also be referred to in the discussion. What I found missing in the article were references to data from the last three years, with a total of 5 articles on the same topic published in 2023. I believe referring to these data would benefit the article's merit.
Overall, the article is worthy of publication in Healthcare. However, the authors should consider the suggestions before publication. Specifically, it is about changing references.
Yours sincerely
Reviewer
